# Psychometric Adaptation and Validity of the Resistance to Peer Influence Scale Among Young Chinese Drivers and Its Links with Peer Pressure and Risky Driving Behaviours

**DOI:** 10.3390/bs15091237

**Published:** 2025-09-11

**Authors:** Wenchengxu Li, Jiahong Liu, Yuxi Wang, Long Sun

**Affiliations:** School of Psychology, Liaoning Normal University, Dalian 116029, China; lwcx1234567890@163.com (W.L.); 15703548498@163.com (J.L.); wangyxi22@163.com (Y.W.)

**Keywords:** peer pressure, risky driving, resistance to peer pressure, young drivers

## Abstract

Risky driving behaviour is closely related to traffic accidents, and the tendency to engage in such behaviour is related to a driver’s ability to resist peer pressure. However, to our knowledge, the relationship between risky driving behaviour and the ability to resist peer pressure among young drivers in China remains unexplored. This study aimed to translate and adapt the Resistance to Peer Influence (RPI) Scale to Chinese drivers and examine whether RPI can moderate the influence of peer pressure on risky driving behaviours. A total of 269 drivers were recruited for this research. These drivers completed the Safe Driving Climate among Friends (SDCaF) Scale, the Peer Pressure on Risky Driving Scale (PPRDS), the RPI Scale and a scale that measured risky driving behaviours. The Chinese version of the RPI scale consists of 10 items and has acceptable reliability. The significant correlations observed among the RPI scale, the SDCaF, the PPRDS and risky driving behaviour indicate that the convergent and discriminant validity of the RPI scale is satisfactory. RPI, friend pressure and shared commitment explained 16.5% of the variance in risky driving behaviour, whereas RPI and risk-encouraging direct peer pressure explained 15.8% of this variance. RPI moderated the relationship between shared commitment and risky driving behaviour. Lower levels of shared commitment combined with low RPI were linked to higher levels of risky driving. RPI also moderated the relationship between risk-encouraging direct peer pressure and risky driving behaviour. Higher levels of risk-encouraging peer pressure were associated with more risky driving regardless of the level of RPI. RPI has acceptable internal consistency and validity and has the potential to serve as a valid tool for assessing and training young drivers in China.

## 1. Introduction

According to data provided by the [8] ([8]), young drivers between the ages of 18 and 25 years represent 10.9% of all drivers. However, they are responsible for 14.07% of all road traffic fatalities in China, resulting in 34,206 injuries and 8048 deaths. Previous studies have consistently reported that risky driving behaviours among young drivers are significantly associated with increased crashes (e.g., [23]). Peer influence or pressure has long been the focus of studies on the many factors that influence risky driving behaviour.

### 1.1. Peer Pressure, Resistance to Peer Pressure and Risky Driving Behaviour

Peer pressure refers to situations in which individuals are motivated to act and think in certain ways because they have been urged, encouraged, or pressured by a peer to do so ([11]). Because they lack driving experience and seek social recognition from peers, young drivers may struggle to accurately assess driving risk. They may also find it difficult to resist negative peer influence while driving ([3]). Studies on this topic have typically reported that negative peer pressure may contribute to higher levels of risky driving behaviour ([3]; [4]; [23]), whereas positive peer pressure is associated with lower levels of risky driving behaviour ([1]; [6]; [19]). Despite the importance of peer pressure, studies have reported that high levels of resistance to peer pressure are associated with fewer unsafe driving behaviours ([10]) and risky decisions ([15]). [15] ([15]) examined the risky decision-making exhibited by participants between the ages of 18 and 75 years in six daily life scenarios and reported that lower levels of resistance to peer influence were related to higher levels of risky decisions, especially when the focal individuals were accompanied by a friend (as opposed to driving alone).

Many instruments have been developed to measure the influence of both peer pressure and resistance to peer pressure on the driving behaviour exhibited by young drivers ([4]; [16]; [19]). The most commonly used scales include the Safe Driving Climate among Friends (SDCaF) Scale, the Peer Pressure on Risky Driving Scale (PPRDS), and the Resistance to Peer Influence (RPI) Scale. The SDCaF was developed by [4] ([4]) and consists of four factors: friend pressure, social cost, communication, and shared commitment. This scale has been translated into Chinese and Spanish (e.g., [10]; [23]) and has been widely used to measure the driving safety climate among peers and its influence on the driving behaviour of young drivers. Studies have reported that the shared commitment and friend pressure factors included in the SDCaF are positively associated with risky driving behaviours ([4]; [23]).

The PPRDS, which was developed by [19] ([19]), originally focused on a sample of drivers from Argentina. The scale consists of three factors: risk-encouraging direct peer pressure, risk-discouraging direct peer pressure, and indirect peer pressure. Risk-encouraging direct pressure refers to pressure that is exerted directly on an individual by that individual’s peers with the goal of encouraging the individual to engage in high-risk behaviour. Indirect pressure refers to the pressure that an individual experiences as a result of observing others’ behaviour ([19]). The PPRDS has been translated into Turkish and Chinese ([1]; [6]). Unsafe pressure from peers—whether direct or indirect—has been linked to an increased tendency to engage in risky driving behaviour (e.g., [6]).

The RPI scale, developed by [16] ([16]), was designed to measure adolescents’ ability to resist the influence of peer pressure. The 10-item RPI scale has a unidimensional factor structure. Each item contained two neutral sentences, an approach that aims to mitigate social desirability bias. The participants must decide which of the two types of persons described in these sentences they are likely to be in most cases. [16] ([16]) reported that resistance to peer pressure increases linearly among adolescents between the ages of 14 and 18 years but not among young adults between the ages of 18 and 30 years. However, some Chinese studies have reported that as age increases, young Chinese drivers between the ages of 18 and 25 years tend to face more indirect peer pressure ([6]) and engage in riskier driving behaviour ([20]). To our knowledge, no studies have examined the relationship between young drivers’ resistance to peer pressure and their risky driving behaviour in China.

[10] ([10]) adapted the RPI scale to the context of Spanish drivers and reported that friend pressure and social cost factors of the SDCaF were negatively correlated with RPI, whereas the shared commitment and communication factors of the SDCaF were positively correlated with RPI. These authors also reported that RPI was negatively correlated with risky, dissociative and angry driving styles, suggesting that individuals who exhibit greater resistance to peer influence are less likely to engage in unsafe driving behaviours. However, the question of whether young drivers’ resistance to peer pressure moderates the effects of peer pressure on their risky driving behaviour remains unanswered.

Few studies have examined the mediating role of resistance to peer pressure in the relationship between risky driving behaviour and decision-making ([14]; [15]). [14] ([14]) examined the mediating role played by resistance to peer pressure in the relationship between parental bonds and risky driving among adolescents between the ages of 16 and 20 years and reported that only the mother–adolescent relationship positively shaped adolescents’ ability to resist peer influence, thereby reducing their risky driving. [20] ([20]) reported that the relationship between peer pressure and risky driving behaviour is moderated by both the family climate for road safety and the safety attitudes exhibited by young drivers. Given the close link between resistance to peer pressure and safety attitudes (e.g., [7]) and the considerable influence of peer groups on young drivers (e.g., [1]; [6]; [10]; [19]), this study examines whether resistance to peer influence moderates the effect of peer pressure on risky driving behaviour.

### 1.2. Aim of This Study

The main purpose of this research is to adapt the RPI scale to young Chinese drivers. The second purpose of this study is to examine the moderating role played by RPI in the relationship between the factors of the SDCaF (or of the PPRDS) and risky driving behaviours. Two hypotheses are proposed.

**Hypothesis** **1.**
*In line with the finding of [10] ([10]) showing significant associations between RPI, positive and negative peer influence and driving styles, this study predicted that RPI is positively correlated with communication, shared commitment and indirect peer pressure and negatively correlated with risky driving behaviour, friend pressure, and risk-encouraging direct and indirect peer pressure.*


**Hypothesis** **2.**
*RPI moderates the relationships between shared commitment and both friend pressure (or risk-encouraging direct peer pressure) and risky driving behaviour. China has a strong collectivist culture, and individuals value comments from others, which makes young people more likely to adjust their actions in line with group commitments and peer expectations. Hence, it is possible that young drivers could perform more risky driving behaviours if they cannot resist risk-taking advice from their peers while driving.*


## 2. Methods

### 2.1. Participants and Procedure

Three hundred and twenty young drivers with valid driving licences were recruited from Shanghai (southeast China), Dalian (northeast China) and Chongqing (southwest China), which are three cities that are geographically representative of China. Two hundred and eighty-one drivers agreed to participate in this research after they were informed of the purpose of this study. The researchers ultimately collected valid data from 269 drivers. The participants were instructed to complete the paper-and-pencil survey described in Section 2.2 within 30 min. Ads were placed at the entrance of one Walmart in each city, and shoppers who were interested in this study completed the survey. The participants first completed a demographic questionnaire that aimed to collect information such as their sex. Each participant received 3 RMB following completion of the survey. This study used G-power software version 3.1.9.2 to select the appropriate minimum sample size. Considering the effect size of 0.20, alpha of 0.05, and the number of predictors, a minimum total sample size of 105 respondents was required for this study.

The final sample consisted of 133 males (49.4%) and 136 females (50.6%). The participants’ ages ranged from 18 to 25 years (*M* = 22.10, *SD* = 2.21), and their driving experience ranged from 0.5 to 3 years (*M* = 1.38, *SD* = 0.77). With respect to their educational backgrounds, 43.1% of the participants had a college education, 28.6% had a high school education, and 28.3% had a middle school education. In terms of their driving frequency, 12.6% of the participants drove once every half-month, 19.0% drove 1 to 3 times per week, 33.1% drove 4 to 6 times per week, and 35.3% drove every day. A total of 61.3% of the participants had driven with a peer/friend during the previous week.

### 2.2. Measures

#### 2.2.1. Safe Driving Climate Among Friends (SDCaF) Scale

The Chinese version of the SDCaF consists of 4 factors: shared commitment (4 items, I feel that my friends are proud of me when I drive carefully, Cronbach’s α = 0.91), communication (5 items, My friends and I talk freely about how each of us drives, α = 0.70), friend pressure (4 items, If my friends ask me to do something when I’m driving, it’s hard for me to refuse, α = 0.71), and social costs (5 items, When my friends are in the car, it makes me feel uncomfortable, α = 0.88) ([23]). These items were scored on a 5-point Likert scale ranging from “1 = completely disagree” to “5 = strongly agree”.

#### 2.2.2. Peer Pressure on Risky Driving Scale (PPRDS)

The Chinese version of the PPRDS consists of 3 factors: risk-encouraging direct peer pressure (6 items, When I’m driving, my friend urges me to drive faster, α = 0.87), risk-discouraging direct peer pressure (5 items, When I’m driving, my friend discourages me from using my mobile phone, α = 0.80), and indirect peer pressure (9 items, My friend would approve of me not stopping at a STOP sign, α = 0.94) ([6]). These items were scored on a 5-point Likert scale ranging from “1 = completely disagree” to “5 = strongly agree”.

#### 2.2.3. Resistance to Peer Influence (RPI) Scale

The RPI scale contains 10 items and is unidimensional ([16]). Each item contains two opposing descriptions: “partially true” and “completely true”. The participants are asked to choose one of the two options to indicate the degree to which the items accurately describe their corresponding behaviour.

Following the translation/back-translation procedure, the original RPI scale was first translated into Chinese by one bilingual researcher, after which another bilingual researcher translated the Chinese version of this scale into English. After the two versions were compared, one policeman and one young driver (both male) were invited to assess each item in terms of linguistic accuracy and frequency of occurrence in the Chinese driving context. High levels of interrater reliability were achieved for all 10 items.

#### 2.2.4. Risky Driving Behaviour

The six items pertaining to risky driving behaviour used in this study focused on frequently reported risky driving behaviours among young drivers in China ([6]; [23]). These items included running red lights, tailgating, speeding, illegal overtaking, driving without a seatbelt, and drunk driving. The items were scored on a 5-point Likert scale ranging from ‘‘never (1)” to ‘‘very often (5)”. Higher total scores indicate a stronger tendency to engage in these risky behaviours. The reliability (α) for this measure was 0.70.

### 2.3. Data Analysis

SPSS 23.0 and Mplus Editor software version 8.3 were used to analyse the data collected for this research. First, a confirmatory factor analysis (CFA) was conducted to examine the factorial structure of the RPI scale. Second, the correlations among the SDCaF, the PPRDS, the RPI scale and risky driving behaviour were analysed to assess its discriminant validity and convergent validity. Third, the moderating role of RPI on the relationships between the factors of the SDCaF and risky driving behaviour and between the factors of the PPRDS and risky driving behaviour were analysed by hierarchical regression analyses. The significant interaction terms between RPI and the factors of the SDCaF (or of the PPRDS) were analysed via a simple slope analysis conducted on the basis of the procedure proposed by [5] ([5]). The assumptions of normality and all the assumptions for parametric testing were met in this study.

## 3. Results

### 3.1. Factor Structure of the RPI Scale: CFA

To examine the factor structure of the RPI scale, weighted least squares means and variance adjustment methods were used as estimation methods. The results indicated that the model fit was acceptable, χ^2^(35) = 87.146, *p* < 0.01, CFI = 0.985, TLI = 0.981, SRMR = 0.031, RMSEA = 0.074 (90% CI: 0.054–0.094). The normalised factor loadings and item descriptions are presented in the Appendix A.

### 3.2. Correlations Among the SDCaF, the PPRDS, the RPI and Risky Driving Behaviour

Reliability analysis revealed that the internal consistency reliability of the RPI scale was 0.899. The correlations among the SDCaF, the PPRDS, the RPI scale and risky driving behaviour were calculated and are presented in Table 1.

The data in Table 1 indicate that RPI scores are positively associated with communication and shared commitment, as well as risk-discouraging direct peer pressure, but negatively associated with friend pressure, social cost, risk-encouraging direct, indirect peer pressure and risky driving behaviour. The results indicate that the convergent and discriminant validity of the RPI are acceptable.

### 3.3. RPI and Demographic Variables

The effects of demographic variables on RPI were examined by conducting a multivariate analysis of covariance (MANCOVA). The results revealed that the effects of sex, *F*(1, 263) = 18.17, *p* < 0.01, were significant, whereas the effects of age, *F*(1, 263) = 0.03, *p* > 0.01; driving experience, *F*(1, 263) = 0.02, *p* > 0.05; driving frequency, *F*(1, 263) = 0.15, *p* > 0.05; and education, *F*(1, 263) = 0.28, *p* > 0.05, were not significant. The independent samples *t* test results revealed that males (*M* = 2.46, *SD* = 0.76) obtained lower scores than females did (*M* = 2.82, *SD* = 0.57) in terms of their resistance to peer pressure, *t* = −4.28, *p* < 0.01, *Cohen’s d* = −0.52.

### 3.4. Predictive Factors for Risky Driving Behaviours

To assess multicollinearity, we computed variance inflation factors (VIFs). All the predictors had VIF values < 2.62, which are within commonly recommended thresholds (e.g., VIF < 3), indicating no serious multicollinearity.

To investigate the influence of RPI and the factors of the SDCaF (or of the PPRDS) on risky driving behaviour, a hierarchical regression analysis was conducted, in which RPI and the factors of the SDCaF (or of the PPRDS) were included as independent variables and risky driving behaviour was included as the dependent variable while controlling for demographic factors in the first step. The interaction terms between RPI and the factors of the SDCaF (or the interaction terms between RPI and the factors of the PPRDS) were entered into the regression (only the interaction whose effect reaches significance is shown in the tables). The results are presented in Table 2 (Table 3 for the PPRDS).

Table 2 indicates that RPI, friend pressure and shared commitment significantly predicted risky driving, explaining 16.5% of the variance in this factor. The interaction between RPI and shared commitment explained an additional 1% of the variance. The data in Table 3 indicate that RPI and risk-encouraging direct peer pressure significantly predicted risky driving and explained 15.8% of the variance in this factor. The interaction between RPI and risk-encouraging direct peer pressure explained an additional 1% of the variance.

### 3.5. Analysis of the Moderating Effects

The PROCESS Model 1, developed by [5] ([5]), was used to examine the effects of the significant interaction terms between RPI and shared commitment and the corresponding interaction term between RPI and risk-encouraging direct on risky driving behaviour. The results are illustrated in Figure 1 and Figure 2.

The data in Figure 1 indicate that higher scores for shared commitment are associated with less risky driving behaviour when the level of RPI is low (*b* = −0.21, *t* = −3.94, *p* < 0.01); however, this factor does not have a positive effect when the level of RPI is high (*b* = −0.05, *t* = −0.90, *p* = 0.37).

The data in Figure 2 indicate that higher scores for risk-encouraging direct peer pressure are associated with higher levels of risky driving behaviours when the level of RPI is low (*b* = 0.34, *t* = 5.50, *p* < 0.01) or high (*b* = 0.15, *t* = 2.10, *p* < 0.05).

## 4. Discussion

As part of this study, the RPI scale was adapted to suit the road traffic safety characteristics and driver habits that are prevalent in China. Our results revealed that the reliability of the RPI scale is acceptable and that the items in this scale exhibit cross-cultural stability ([16]; [22]). The validity of the RPI scale was supported by the significant associations among the RPI, risky driving behaviour and peer pressure, as measured by the SDCaF and the PPRDS.

In line with the findings of previous studies (e.g., [13]), a significant sex difference was observed in individuals’ RPI scores, in which females obtained higher RPI scores than males did. However, this study did not reveal any significant associations between age and RPI. Although [18] ([18]) reported that adolescents’ ability to resist peer pressure increases with age, the impact of age is evident only during adolescence. Other studies have reported that age does not affect individuals’ ability to resist peer influence (e.g., [2]).

This study revealed that RPI is significantly correlated with peer influence, as measured by the SDCaF. In line with the findings of previous researchers ([10]; [14]; [15]), RPI is positively correlated with shared commitment but negatively correlated with friend pressure and social cost. Moreover, this study demonstrated that RPI is significantly associated with both direct and indirect peer pressure while driving. These results support Hypothesis 1 and provide evidence to support the discriminant and convergent validity of RPI. The regression analysis conducted for this research revealed that the shared commitment and friend pressure factors of the SDCaF and the risk-encouraging direct peer pressure factor of the PPRDS significantly predicted risky driving behaviour. These results replicated the findings of previous studies that reported that shared commitment and direct peer pressure associated with risk-taking are closely related to risky driving behaviour among young drivers (e.g., [6]; [19]).

In line with Hypothesis 2, this study initially contributes to the literature by revealing that RPI moderates the effects of risk-encouraging direct peer pressure on risky driving behaviour. Specifically, higher scores for risk-encouraging direct peer pressure were revealed to be associated with higher levels of risky driving behaviour regardless of the individual’s ability to resist peer pressure. The results revealed that risk-encouraging direct peer pressure could influences risky driving behaviour despite young drivers’ desire to resist peer pressure while driving. Notably, these results suggest that the moderating effect of RPI is less pronounced when peers strongly encourage risk-taking. In such situations, the protective role of RPI may be limited, especially when individuals are under strong social pressure to engage in risky driving. Collectivist values stress harmony and conformity and declining peers’ direct requests is often seen as undermining group cohesion or causing a loss of face. As a result, young drivers may comply even when they generally show strong resistance to peer influence. In this way, cultural expectations of harmony and face-saving can weaken the effect of RPI against risk-encouraging direct peer pressure. [12] ([12]) reported that when drivers experience direct pressure from peers to take risks, they tend to drive dangerously and to engage in higher levels of risky driving behaviour, such as speeding. Similarly, [1] ([1]) reported that risk-encouraging direct pressure has a positive effect on self-reported risky driving behaviour. This effect remains evident even when these drivers are willing to resist the influence of their peers. Notably, according to the findings of this study, the tendency of young drivers to engage in risky driving behaviour is significantly lower when their RPI is higher (as opposed to lower).

This study also examined the moderating role played by RPI in the relationship between peer influence (i.e., the shared commitment factor of the SDCaF) and risky driving behaviour. Higher scores for shared commitment and RPI (as opposed to lower levels of RPI) can jointly lead to significant reductions in risky driving behaviour. The results suggested that efforts to cultivate a safe driving climate among peers and enhance their ability to resist peer pressure could play protective roles with respect to young drivers’ risky driving behaviour. [20] ([20]) reported that a strong sense of collective responsibility (i.e., shared commitment) among young drivers and their friends can decrease the tendency to engage in extreme behaviours while driving, even if their parents spend less time promoting safe driving education and practices in their daily lives.

## 5. Limitations

This study has several limitations. First, the relationships among the variables included in this investigation might be influenced by the self-reported data. Future researchers should obtain more objective data on the basis of driving simulators or naturalistic observations. However, several studies have reported that social desirability biases in drivers’ self-report are not as problematic as may be expected (e.g., [17]). Another limitation of this research is that the size of the sample investigated is relatively small, and the participants are mainly recruited from urban areas. This could limit the generalizability of the findings to the entire population of young drivers in China. Future research could emphasise replication of the findings with larger and more diverse groups. Third, this study focuses on six commonly reported risky driving behaviours among young drivers that are closely related to crash risk (e.g., running red lights). However, this study treats these six driving behaviours as a whole rather than treating each as a separate dependent variable. Future researchers should explore the specific effects of peer pressure and RPI on each of these behaviours in further detail. Finally, a cross-sectional design was used, and follow-up studies are recommended to examine the relationships between the RPI, risky driving behaviour and crash risk.

## 6. Implications

This study also has several implications. First, this research provides a reliable scale that can be used in this context, thereby offering a potential method of measuring or evaluating young drivers’ ability to resist peer influence while driving. Second, this study provides a multidimensional framework for young drivers’ risky driving behaviour that includes both peer pressure and the ability to resist such peer pressure. Third, the moderating effect of RPI on the relationship between peer pressure and risky driving behaviour also provides insights that can help practitioners and policy makers develop training programmes aimed at enhancing young drivers’ ability to resist peer pressure. For example, interventions involving driving simulators, in which both young drivers and their peers participate, might improve these drivers’ visual search patterns and ability to respond to potential hazards in a timely manner ([9]). Other approaches, such as interactive safety quizzes ([21]), could also help young drivers become aware of the danger posed by their peers while driving, thereby enhancing their safety attitudes.

## 7. Conclusions

This study adapted the RPI scale to the context of young Chinese drivers and confirmed its acceptable psychometric properties. The significant correlations observed between RPI and the factors of the SDCaF and the PPRDS support the convergent validity of this scale and highlight the influence of peer influence on risky driving behaviours. This study also highlights that RPI moderates the effects of shared commitment and risk-encouraging direct peer pressure on risky driving. The findings of this research not only enrich our understanding of peer-related influences on risky driving behaviours among young drivers in China but also provide new insights that can support the development of interventions aimed at reducing peer-induced risky driving.

## Figures and Tables

**Figure 1 behavsci-15-01237-f001:**
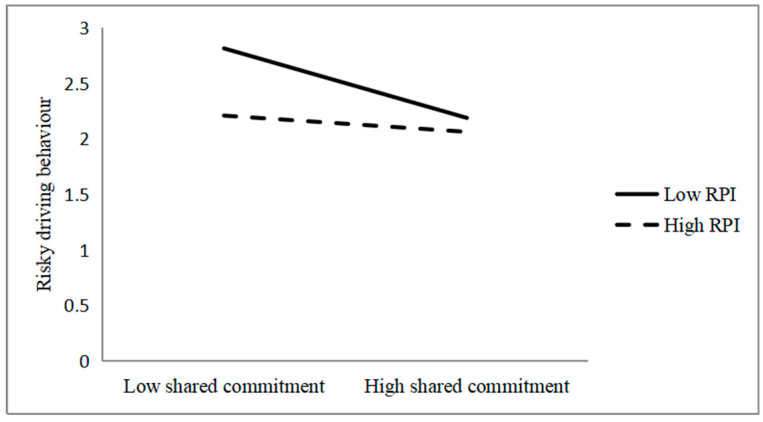
Interaction between shared commitment and RPI in predicting risky driving behaviour.

**Figure 2 behavsci-15-01237-f002:**
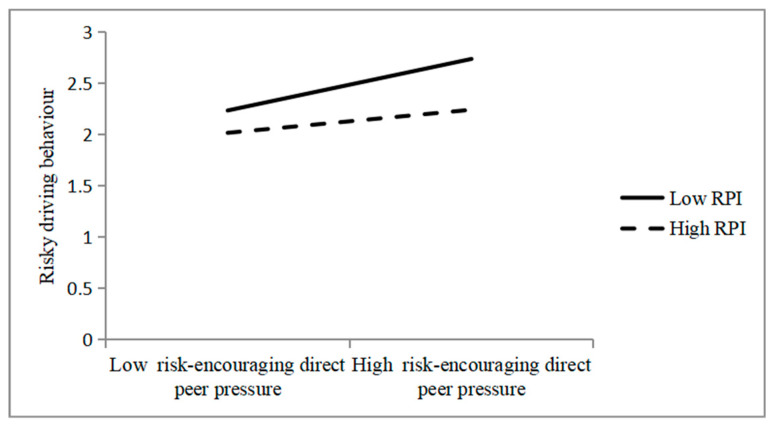
Interaction between risk-encouraging direct peer pressure and RPI in predicting risky driving behaviour.

**Table 1 behavsci-15-01237-t001:** Correlations among the variables included in the study (*n* = 269).

Item	1	2	3	4	5	6	7	8	9
Friends pressure (1)	1								
Social cost (2)	0.51 **	1							
Communication (3)	−0.03	−0.05	1						
Shared Commitment (4)	−0.28 **	−0.38 **	0.62 **	1					
Risk-encouraging direct (5)	0.59 **	0.58 **	−0.13 *	−0.46 **	1				
Risk-discouraging direct (6)	−0.27 **	−0.22 **	0.31 **	0.36 **	−0.12	1			
Indirect peer pressure (7)	0.49 **	0.58 **	−0.19 **	−0.41 **	0.68 **	−0.18 **	1		
RPI (8)	−0.32 **	−0.20 **	0.12 **	0.27 **	−0.22 **	0.22 **	−0.23 **	1	
Risky driving behaviour (9)	0.36 **	0.24 **	−0.08	−0.28 **	0.35 **	−0.15 *	0.35 **	−0.34 **	1

Note: * *p* < 0.05, ** *p* < 0.01.

**Table 2 behavsci-15-01237-t002:** Regression coefficients (standardised beta weights) for predicting risky driving behaviour (*n* = 269).

Variables	*B*	*t*
RPI	−0.15	−2.19 **
Friend pressure	0.23	−3.53 **
Social cost	0.01	0.10
Communication	0.06	0.85
Shared commitment	−0.17	−2.02 *
Δ*R*^2^	0.165	
RPI × shared commitment	0.14	2.07 *
Δ*R*^2^	0.01	

Note: * *p* < 0.05, ** *p* < 0.01.

**Table 3 behavsci-15-01237-t003:** Regression coefficients (standardised beta weights) for predicting risky driving behaviour (*n* = 269).

Variables	*B*	*t*
RPI	−0.20	−3.05 **
Risk-encouraging direct	0.16	2.03 *
Risk-discouraging direct	−0.06	−1.04
Indirect peer pressure	0.11	1.42
Δ*R*^2^	0.158	
RPI × risk-encouraging direct	−0.13	−2.03 *
Δ*R*^2^	0.01	

Note: * *p* < 0.05, ** *p* < 0.01.

## Data Availability

The original data presented in the study are openly available in [FigShare] at [10.6084/m9.figshare.29815862].

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
