# Peer review of "Psychometric Adaptation and Validity of the Resistance to Peer Influence Scale Among Young Chinese Drivers and Its Links with Peer Pressure and Risky Driving Behaviours"

_behavsci, 2025, doi:10.3390/bs15091237_

Round 1
Reviewer 1 Report
Comments and Suggestions for Authors
This paper tackles an important and relatively underexplored topic: how peer influence shapes risky driving among young Chinese drivers, and how the Resistance to Peer Influence (RPI) Scale can be adapted for this group. The research is timely and highly relevant to traffic safety, offering insights that could inform interventions and policy. Overall, the paper is clearly organised, with solid methods and analyses.
That said, there are several areas where the manuscript could be strengthened:
Literature Review and Background
-
The introduction currently cites a very large number of studies, some of which overlap in content. Streamlining the review to focus on the most directly relevant work would make the argument stronger.
-
The section on adapting the RPI scale would benefit from more discussion of cultural factors. Beyond translation, how were items judged to be appropriate for the Chinese driving context?
Methods
-
The recruitment process needs more detail. For instance, were participants recruited through driving schools, universities, or social media? Clarifying this helps readers judge how representative the sample is.
-
The reliance on Harman’s single-factor test for common method bias is a limitation. It would be better to acknowledge its weaknesses and, if possible, add other checks—or at least explain why it is sufficient in this case.
Results and Interpretation
-
The moderation analysis shows that peer pressure encouraging risk predicts risky driving regardless of RPI level. This weakens the claim that RPI moderates the relationship. The discussion should reflect this and note the limited role of moderation.
-
Tables should specify whether coefficients are standardised or unstandardized.
Discussion and Conclusions
-
Be careful not to generalise too broadly, given the relatively small, urban sample. Future research could emphasise replication with larger and more diverse groups.
-
The cross-sectional design is another limitation worth acknowledging, since it prevents firm conclusions about causality.
Language and Style
-
The writing is clear overall, but some sentences—especially in the abstract and introduction—are too long and could be simplified for readability.
-
Use consistent terminology throughout. The paper switches between “risky driving” and “reckless driving,” which could confuse readers. Choose one term and define it clearly.
Overall Assessment
This study has strong potential and makes a valuable contribution. With revisions to improve clarity, methodological transparency, and interpretation, the paper would be much stronger.
Author Response
A response letter is uploaded.

Reviewer 2 Report
Comments and Suggestions for Authors
See attached file.

Author Response
A response letter is uploaded

Reviewer 3 Report
Comments and Suggestions for Authors
The Resistance to Peer Influence (RPI) is a questionnaire with ten items. Each item consists of two opposing descriptions. Of each pair, the participant has to choose the description that fits her or him best and to indicate if what is stated in this description is ‘partially true’ or ‘completely true’. The RPI has one scale. The researchers wanted to know if the RPI has the psychometric properties for use in China. They also wanted to know if there was an association between the scores on the RPI and scores on other instruments to measure aspects of peer pressure (the Safe Driving Climate among Friends (SDCaF) and the Peer Pressure on Risky Driving Scale (PPRD), and self-reported risky driving behavior of young novice drivers. This, to check if the RPI has convergent and discriminant validity. The researchers also wanted to know if RPI moderates the relationship between shared commitment (a scale of the SDCaF) and self-reported risky driving behavior and friend pressure (scales of the PPRD) and self-reported risky driving behavior. To this end 269 young drivers (18-25 years of age) completed the RPI, the SDCaF, the PPRD, the risky driving behavior questionnaire, and a questionnaire about demographics. The model fit indices of the CFA were acceptable, and the alpha was high, indicating that the RPI can be used in China. A correlation matrix revealed that there were quite strong correlations between the RPI, self-reported risky driving behavior, and some of the scales of the SDCaF and the PPRD, indicating that the RI has some convergent and discriminant validity. A hierarchical multiple regression with risky driving behavior as dependent variable and demographics entered in the first step, the RPI and the scales of the SDCaF entered in the second step, and the interactions of the RPI and the scales of the SDCaF entered in the third step, revealed that after controlling for demographics, the RPI, ‘Friend Pressure’, and ‘Shared commitment’ were significant predictors of self-reported risky driving behavior. Of the interactions in the third step, only RPI x ‘Shared commitment’ was a significant predictor. A second hierarchical multiple regression with risky driving behavior as dependent variable and demographics entered in the first step, the RPI and the scales of the PPRD entered in the second step, and the interactions of the RPI and the scales of the PPRD entered in the third step, revealed that after controlling for demographics the RPI and ‘Risk-encouraging direct’ were significant predictors of self-reported risky driving behavior. Of the interactions in the third step, only the interaction RPI x ‘Risk-encouraging direct’ was a significant predictor of self-reported risky driving behavior. To examine the moderating effect of RPI on the relationship between ‘Shared commitment’ and self-reported risky driving behavior and to examine the moderating effect of RPI on the relationship ‘Risk-encouraging direct’ and self-reported risky driving behavior (the two significant interactions of the hierarchical regressions), two Hayes PROCESS Model 1 were conducted. The first revealed that decline in risky driving behavior between those with low ‘Shared commitment’ and in those with high scores for ‘Shared commitment’ was rather steep for those with low RPI-scores but was less steep for those with high scores for RPI. The second revealed that the risky driving score was higher for those with high scores for ‘Risk-encouragement direct’ than for those with low scores for ‘Risk-encouragement direct’ but that the moderating effect of RPI did not differ much between the two.
The article is well written, well structured and the data is treated correctly. I only have a few minor comments
Comments
In the last line of the abstract (line 32) the authors write that the RPI can serve as a valid tool in China. This can only be the case when the sample of the study is representative for young drivers in China (regarding for instance sex and education level). Do the researchers have an indication that the sample was representative?
Section 2 Methods). It is not mentioned in this section if the study was approved by an ethics committee.
Section 2.1. (Participants and procedure). It is not mentioned in this section how participants were recruited, and it is also not mentioned where the participants completed the questionnaires.
Section 2.2. (Measures). For those who are not familiar with the SDCaF it is not always clear what the scales of the tests indicate. What is ‘social cost’? What is ‘shared commitment’? What is ‘communication’. Could the researchers provide an example item of each scale? I would also like to see example items of the scales of the PPRDS and an example of one item of the RPI.
Section 2.4. (Data Analysis). The researchers applied parametric tests. However, they do not write if the assumptions for parametric testing were met.
Section 3.4. (Predictive factors for risky driving behaviours), lines 228-229. Iwould write: “…while controlling for the demographic factors in the first step.”
Section 3.4. (Predictive factors for risky driving behaviours), lines 230-231. The authors write: “…were entered into the regression in the third step (via the stepwise method).” The text in brackets confuses me. As far as I know, the predictors in a step are entered together (with the enter method and not with one of the stepwise methods). If this is so, I hink it is better to delete ‘(via the stepwise method)’. If the stepwise method was really used, please explain why?
Table 2. Shouldn’t the R-squared under ‘RPI x shared commitment’ be ‘Delta R squared?
Table 3. Shouldn’t the R-squared under ‘RPI x risk-encouraging direct ’ be ‘Delta R squared?
Author Response
A response letter is uploaded
